# The OWL Screening Tool—A Protocol for Holistic Pediatric Lifestyle Assessment

**DOI:** 10.3390/healthcare13212731

**Published:** 2025-10-28

**Authors:** Alina Auffermann, Wolfgang Auffermann

**Affiliations:** 1German Clinic, Al Razi Medical Complex, Dubai Healthcare City, Dubai 505055, United Arab Emirates; 2Dr Sulaiman Al Habib Hospital, Dubai Healthcare City, Dubai 505005, United Arab Emirates

**Keywords:** health risk, screening, tool, lifestyle, pediatric assessment

## Abstract

**Background/Objectives**: The identification of health risk factors in children should rely not only on body mass index but also on modifiable lifestyle behaviors. Early screening for physical inactivity, poor nutrition, inadequate sleep, and chronic stress is crucial for effective preventive healthcare. The aim of this project was to develop the OWL screening tool, a protocol for the holistic assessment of key lifestyle risk factors in children aged 6–12. **Methods/Rationale**: The OWL tool was developed by integrating evidence-based recommendations from major health societies (WHO, EFSA, the National Sleep Foundation, and the Pediatric Endocrine Society), incorporating psychological principles, and adapting validated components from existing pediatric screening instruments. Its design prioritizes flexibility for use across various age groups and settings. The development process resulted in the 20-item OWL questionnaire, structured into four lifestyle domains: nutrition, physical activity, sleep, and stress management. Each item is a closed-ended question requiring a dichotomous (yes/no) response. One point is awarded for each health-promoting behavior endorsed, yielding a total possible score of 20. The tool is suitable for self-report by older children, parent-report for younger children, or clinician-administered review. **Conclusions**: By integrating sleep and stress management with traditional lifestyle factors, the OWL screening tool offers a highly relevant approach to pediatric preventive care. The findings presented here should be interpreted as a proof-of-concept, and the tool is not yet ready for clinical implementation without further rigorous evaluation.

## 1. Introduction

Given the high rates of physical inactivity and overweight among children as they age, there is growing interest in health promotion [1,2]. Furthermore, promoting a healthy lifestyle from a young age appears essential for developing long-term positive behaviors [3,4]. Children are aware of some risks associated with an unhealthy lifestyle (e.g., being overweight or eating an unhealthy diet), but they also recognize several health-related benefits. These include developing social skills, having fun, making friends, maintaining concentration, and feeling happy. Mental health was rated as ‘very important’ by more than 70% of children [2] but has not been evaluated in obesity prevention studies [5]. Since previous research has shown that a holistic approach targeting both physical and mental health has the highest success rate in prevention [6], multidisciplinary interventions may therefore hold significant potential for enhancing children’s health, as well as their academic and cognitive abilities [1,4,7].

Previous studies on the relationship between familial lifestyle challenges (e.g., lack of sleep, unhealthy diet, insufficient exercise) and stress management have demonstrated how these factors interact to influence overall family well-being [3,8]. For instance, interventions such as consistent bedtime routines and screen-time limits have been shown to improve sleep and reduce family stress. Conversely, stress often triggers emotional eating of high sugar foods, which worsens both mental and physical health [3]. Furthermore, excessive screen time disrupts family interaction and sleep; this contrasts with the benefits of mindfulness practices like family meditation or breathing exercises [8,9].

Research has also revealed bidirectional relationships between dietary patterns, sleep, physical activity (PA), and sedentary time [10]. For example, longer sleep duration led to less sedentary behavior and increased PA the following day. Conversely, reduced sleep duration and efficiency were associated with the unhealthy dietary patterns. Although these results illustrate clear interactions between these behaviors, further research is needed concrete guidelines. This need is underscored by the World Health Organization (WHO) latest report, which states there is limited evidence on the relationship between PA, prosocial behavior, and sleep in school-age children [11].

Scientific research on school children is inherently challenging, which often prevents studies from examining multiple lifestyle factors—such as PA, nutrition, sleep, and stress management—simultaneously. Currently, there is a lack of such holistic studies in children aged 6–12 that consider all four of these factors [5,12]. While some studies have focused specifically on improving sleep or stress management [9,13,14], others have targeted PA and diet [1].

Findings from the CogniDO study series, which investigated cognitive performance in school children, showed that children’s daily lives reinforce unfavorable behavior patterns [15]. Poorer cognitive performance was correlated with lower PA, poorer sleep quality, and skipping breakfast. Most existing intervention studies have taken a holistic approach by focusing on increasing PA, improving dietary behavior, and reducing sedentary behavior/screen time [5]. The authors of this research further suggest that an easy-to-use tool is needed to classify risks and provide practical approaches for protecting children from health risks.

To date, nutrition-focused lifestyle screening instruments have been used primarily for obesity prevention in school children [5]. The PREDIMED study was the first large randomized controlled trial to address gaps in nutrition research for adults by focusing on whole dietary patterns rather than single nutrients and by measuring clinical endpoints. Its resulting 14-point Mediterranean Diet Adherence Score became a widely used tool to guide dietary intervention [16,17,18]. A corresponding quality index for the Mediterranean diet in children and adolescents, the KIDMED index, was subsequently developed [19]. This practical, questionnaire-based tool for data collection could be expanded into a more holistic screening instrument. Such a tool could then serve as a guide for lifestyle optimization for both parents and children.

Therefore, the proposed framework presented here addresses a scientific need by following a new multifactorial approach that considers the four aforementioned lifestyle factors. It aims to provide a screening template to classify risks, thereby creating the ‘easy-to-use tool’ recommended by others [5] for future studies on child health. Consequently, the project objectives were to develop a new screening instrument that (1) contains questions on four lifestyle factors—PA, nutrition, sleep, and stress management; (2) is practical for use outside scientific context to identify lifestyle-related health risks in children (e.g., obesity); and (3) serves a dual function of preventive screening and guidance for healthy living. The present study introduces the conceptual framework and development protocol for the proposed screening tool; however, it is limited by the absence of empirical validation.

## 2. Materials and Methods

Based on the proposed objective, the research instrument was designated the OWL (Optimize Wisely your Lifestyle) screening tool. This is characteristic of a first-phase development study, which focuses on conceptualization and content development. The reliability and validity of this new instrument will be tested in the future as part of the planned OWL cross-sectional study, which examines health risks in children living in the United Arab Emirates (Registration in progress on ClinicalTrials.gov). The questionnaire is written in the two official languages of the UAE, English, and Arabic. This is intended to ensure that it is linguistically adapted to use in the UAE (Appendix A). All socioeconomic and cultural differences in the population will only be captured during the future validation of the tool.

A literature review was conducted to identify and select core information for evaluation. To determine the content of the screening tool, studies and reviews involving children that employed various methods for assessing health risks as part of preventive strategies (including overweight and obesity) were reviewed (Figure 1).

Table 1 summarizes nine key studies that provided insight into designs successfully used in children’s health research, in accordance with a systematic review [5] and other studies [18,20,21]. The development of the new OWL tool was informed by integrating the strengths of these existing methodologies. As a focus on user-friendliness was paramount, potential screening limitations were minimized.

### 2.1. OWL—Screening

The four lifestyle factors—PA, nutrition, sleep, and stress management—should be considered as an integrated concept within the OWL screening framework. This holistic approach aligns with the ‘Clinical Practice Guideline’ for obesity published by the Society for Pediatric Endocrinology [25]. As the OWL screening is also intended to serve as a guide for developing healthy lifestyle habits, the psychological principles of habit formation were deemed crucial. Given that studies demonstrate approach-oriented goals are significantly more effective than avoidance-oriented goals in sustaining behavior change, the tool employs positive language rather than negative instructions [26,27]. To prioritize simplicity, the tool avoids multiple-choice questions. Instead, it utilizes a closed-ended question format, a method successfully employed in the PREDIMED study [16,17,18] and in the KIDMED study [19]. The dietary screeners used in these studies did not use graded response formats. The response format records the level of agreement (yes/no) with the provided recommendations. Compared to the FLY- Kids questionnaire [21], which uses graded response options, the Owl screening tool was developed based on the PREDIMED study. The selected questions were chosen to balance scientific rigor with practical application, ensuring the findings are both statistically sound and meaningful for real-world implementation. The assessment of lifestyle habits can be conducted through interviews with healthcare providers, or via self-reports or parent reports.

Furthermore, existing research demonstrates that comprehensive screening instruments are often challenging to implement in practice and contribute to significant participant burden [1,23]. To mitigate this, the length of the OWL screening tool was informed by the median number of the items (~22) reported in a review of 41 studies [5]. This decision was further supported by evidence showing high reliability for 20-item questionnaires assessing nutrition, activity, sleep, and screen time in children aged 8–10 years [22]. Consequently, a 20-item, child-friendly instrument was developed for the OWL screening, allowing for a maximum of five questions per behavioral domain. One point is awarded for each positive response, yielding a maximum possible score of 20 points. Although this tool was designed for children aged 6–12, its framework allows for adaptation to other age groups by modifying reference values.

### 2.2. Nutrition

A balanced diet emphasizes variety, favors whole grains over refined grains, prioritizes diverse protein sources, avoids sugary drinks, and recommends water as the primary beverage. Limiting the number of questions on nutrition necessitated a focus on specific topics, as recommended by Krijger et al. [5] and aligned with the NutricheQ study [24]. These core topics are the consumption of fruits, vegetables, protein, and fiber. Accordingly, daily food intake should include five portions of fruits and vegetables, 0.66 g of protein per kg of body weight, and at least 25–30 g of fiber [28,29]. The questionnaire incorporated the Mediterranean diet quality index’s provision for three daily protein-rich meals (e.g., dairy products, pulses, fish, nuts), with special emphasis placed on nuts. This aligns with recommendations from nutrition societies that highlight nuts as a source of protein, essential fatty acids, and fiber [19,30]. Reflecting this, the KIDMED index assigns a positive point for consuming nuts at least two to three times per week. As a key source of fiber, carbohydrate-rich foods (such as rice, pasta, bread, cereals, and other grains) are emphasized in dietary recommendations [29] and were therefore included in the OWL questionnaire.

Consistent with the aim of assessing positive dietary (e.g., water intake) rather than the avoidance of negative behaviors (e.g., sweets or soft drink consumption), a specific question on water consumption was included. This decision was based on two key factors: the established relevance of water consumption for children’s cognitive performance at school [20], and data showing that over 80% of children across many European countries consume less water than the European Food Safety Authority (EFSA) guidelines recommend [31].

### 2.3. Physical Activity

For the PA matrix, the questionnaire was designed to assess daily step count, endurance, strength, coordination (based on WHO recommendations), and outdoor activity to reduce time spent in sedentary behavior (e.g., leisure screen time) [11]. The updated WHO 2020 guidelines provide the following evidence-based recommendations: at least 60 min of daily moderate- to vigorous-intensity PA, muscle- and bone-strengthening activities on at least three days per week, and participation in physical activities that are enjoyable and varied [11]. Oswald et al. [32] defined ‘green time’ as time spent in nature, which is associated with favorable psychological outcomes and can serve as a public health resource for children’s psychological well-being.

In line with the WHO’s “Let’s Get Moving!” campaign, which promotes the integration of walking and cycling into daily life, a target of 10,000 steps per day was included as a preventative measure for mental health [33]. However, controlled studies indicate that continuous walking at moderate intensity generates approximately 3300–3500 steps in 30 min, or 6600–7000 steps in 60 min, for 10–15-year-olds. Furthermore, research suggests that primary school children typically achieve an average of 60 min of at least moderate-intensity PA, corresponding to a total volume of 13,000 to 15,000 steps/day for boys and 11,000 to 12,000 steps/day for girls [34]. A study using accelerometers confirmed that the 10,000-step target is easily achievable for children under 12, as their average step count was approximately 15,000 [20].

### 2.4. Sleep

From the WHO list of critical health outcomes for children and adolescents, sleep duration and quality were selected as important metrics [11]. A consistent bedtime routine is one of the most important factors not only for healthy sleep but also for the overall child development and well-being [8]. Interventions to improve sleep and cognition have shown that consistent bedtimes and reduced screen time before bed improve sleep efficiency. Therefore, the use of various media before falling asleep—ideally, avoided for at least one hour before bedtime –in favor of other activities, in accordance with expert recommendations [35]. Based on National Sleep Foundation guidelines, the optimal sleep duration for children aged 6–12 are 9–11 h per night, and for teenagers, it is 8–10 h per night [36]. Since short sleep duration correlates with a late bedtime, an appropriate bedtime is of great importance for children’s development [37]. A common method suggested by expert is to calculate bedtime based on the wake time required for school (e.g., if a child must wake up at 6.30 a.m., their bedtime should be between 7:30 and 9:30 pm) [38,39]. Less attention is often paid to wake-up time, even though it can be highly relevant to academic performance. As some studies suggest, a shorter time after waking up can lead to breakfast being skipped [10,15]. Breakfast is not only crucial for school children facing cognitive challenges [23,40,41], but, in contrast to mixed results in adults [42], skipping breakfast is associated with an over 40% increased risk of obesity in children and adolescents [10,43]. Therefore, during the pre-selection process for the questionnaire, particular emphasis was placed on wake-up time and breakfast consumption. Questions about sleep quality were also included, as there is a recognized lack of such data in existing studies [44].

### 2.5. Stress Management

The stress management dimension, which includes questions on time with family, evening rituals, and joy/laughter, was incorporated based on evidence that ’healthy home habits’ reduce stress and improve mood in children [3]. This choice was motivated by the conspicuous absence of this stress factor in other lifestyle screening instruments, despite its well-established link to overall health [5]. For example, consistent bedtimes, mindful family interaction (such as meditation or breathing exercises), shared family activities (e.g., walks, yoga), and engaging in hobbies can increase life satisfaction and improve health and well-being [8,9]. The WHO guidelines on sedentary behavior, which is linked to mental health and prosocial development, emphasize limiting children’s recreational screen time to a maximum of two hours per day to encourage more social activities [11]. Corresponding questions on emotional well-being were developed to address areas children find important, such as developing social skills for making friends and the high value they place on ‘having fun’ or ‘feeling happy’ [2]. The WHO committee also recognized the importance of children spending time on quiet activities (e.g., reading, learning, painting, crafts, playing music, solving puzzles) for their development, highlighting benefits for cognitive outcomes [11].

### 2.6. Tool Draft

Drawing on scientific research and recommendations from healthcare providers, including the WHO, EFSA, the National Sleep Foundation, and the Pediatric Endocrine Society [11,25,29,36], an OWL screening questionnaire was developed for school-age children. Its features are summarized in Table 2.

The 20-item instrument, comprising formulated questions, is suitable for implementation in a tabular layout. This design supports clear presentation and simplifies the evaluation of results, as illustrated in Table 3. To improve comprehension, explanatory notes were added in brackets following the questions. The four lifestyle factors are organized into separate sections for clarity (Section A, B, C, and D). Supplementary to the section headings, visual icons depicting the four dimensions (PA, nutrition, sleep, and stress management, icons not shown) have been included to enhance reader engagement. To function as both a scientific tool and a practical daily guide, the tool includes an explained scoring system to raise awareness of health risks. To avoid bias, the scoring system is not explained during the screening process. However, during the review, practitioners should decide on a case-by-case basis whether an explanation would be useful for their specific purpose.

The selection of screening questions for the study warrants further discussion, as it directly impacts the validity and applicability of the findings. Key considerations must include the specificity of the questions, as well as their relevance to the target demographic.

## 3. Discussion

The aim of this project was to describe the development of a novel screening tool designed to identify risk factors affecting the physical and mental health of school children within a preventive context. Unlike the FLY Kids Lifestyle Screening Tool for children aged 1 to 3 years, the OWL screening represents an extension for an older demographic, incorporating evidence-based recommendations from leading professional societies, [11,25,29,36]. To our knowledge, this is the first tool to adopt a holistic approach that encompasses four lifestyle indicators alongside the child’s environment and is designed to be user-friendly in real-word contexts [5]. Although the tool’s psychometric properties, such as reliability and validity, have not yet been empirically tested and thus cannot be evaluated, the twenty goal-oriented questions it contains synthesize the most critical challenges in modern pediatric prevention [3,25,35]. For school-aged children, the school environment and education also play a crucial role in reinforcing a healthy lifestyle [1,20,45]. Consequently, this screening tool has been designed to integrate relevant aspects of previously successful instruments, addressing influences from both the family and school environments.

The entire OWL concept originated from the COGNIDO study series, which examined the cognitive abilities of school children from both short- and long-term perspective in relation to PA (including school sports), nutrition, screen time, sleep, and body mass index. The results of these studies, which are consistent with the findings of the CLASS study involving children of the same age [23], demonstrated that the combined positive influence of multiple beneficial lifestyle habits has a greater impact on body weight and cognitive abilities than any single habit alone [15,20,46]. This finding suggests that future programs promoting healthy lifestyles among children should adopt a holistic approach rather than focusing on isolated aspects. This is particularly relevant because prevention becomes more challenging once the adverse consequences of an unhealthy lifestyle have become apparent [25].

Unlike previous screening instruments that recorded both positive and negative habits (e.g., Fly-Kids screening and KIDMED test), the OWL concept was designed to focus exclusively on positive language and approach-oriented behavioral goals. For instance, it avoids questions about the restriction of behaviors, such as the consumption of soft drinks, sweets, or screen time. This approach offers several advantages and provides a more robust foundation for practical applications. Research shows that behavioral changes in children are significantly more sustainable when goals are framed positively rather than as restrictions [27]. Furthermore, promoting positive lifestyle goals—such as eating more fruit and vegetables and drinking water—naturally leads to a significant reduction in the intake of nutrient-poor snacks and sweetened beverages [47]. Similarly, encouraging more PA and sufficient sleep-in children engaging in fewer sedentary activities and spending less time in front of screens [10]. Therefore, the use of positive descriptions with clearly formulated expectations effectively redirects behavior healthier patterns.

Attention-redirecting strategies, which can increase psychological well-being within the family, are also incorporated into lifestyle interventions. For example, practicing family meditation together helps to manage stress, improve sleep quality and strengthen parent–child relationship [9]. Psychological support for the family is crucial for achieving optimal long-term health outcomes [4]. For this reason, the OWL screening template was designed to serve as a preventive tool for comprehensive education and counselling in medical facilities and schools, extending its utility beyond the scientific community. This approach aims to reach all families with children facing everyday challenges. When combined with motivational interviewing, the tool can help identify specific needs [3,25].

Despite the relative scarcity of intervention research on sleep quality in school-aged children compared to PA and nutrition [44], sleep hygiene was still given equal weighting in the OWL screening tool. Research has shown that a relatively modest sleep reduction of just half an hour of sleep per night can have a statistically significant negative impact on child’s health-related quality of life [48]. According to a recent brain imaging study, children (aged 9–14) with the poorest sleep performance—including the shortest duration, latest bedtime, and highest nighttime heart rate—also exhibited the poorest cognitive performance [37]. Interestingly, European experts recommend Mediterranean diet to promote sleep [44]. Subsequent analyses revealed that its benefits extended to a wide range of conditions, including reduced risk of diabetes, improved metrics of metabolic syndrome, and better cognitive function [16]. Acknowledging the established role of the Mediterranean diet in mitigating lifestyle-related disease risks, the OWL screening incorporates the KIDMED score, an instrument designed for pediatric research [18] to evaluate dietary pattern conducive to metabolic health and restorative sleep.

Finally, it is important to note that the benefits of ‘food as medicine’ are well-established [49], and PA alone is recognized as a treatment for 26 diseases [50]. Supporting families with strategies related to exercise, nutrition, sleep, and well-being would significantly improve a wide range of health issues. Therefore, a key strength of the OWL concept is its potential to inspire the development of broader screening tools for guiding lifestyle optimization in families. The trend of increasing pediatric care complexity underscores the need for early, multidimensional tools like the OWL screener to identify risk trajectories before they lead to higher care dependency. Recent research confirms this, showing that medical complexity (including weight) significantly correlates with nursing care complexity in hospitalized children [51]. By identifying modifiable lifestyle risks upstream, the OWL tool aims to serve as a preventive instrument to mitigate this trajectory, enhancing its long-term impact. A further strength is the tool’s flexibility, as it can be adapted for other age groups by adjusting reference values.

The primary limitations of the tool are the current lack of formal validity testing. While its development was informed by established clinical frameworks and expert input, its psychometric properties—including reliability, criterion validity, and predictive validity—remain unassessed. Consequently, its findings cannot yet be considered evidence-based for clinical practice or wider implementation. Future research must include rigorous validation studies in diverse pediatric populations to confirm its accuracy and clinical utility. Another limiting factor is a restricted question set (up to 20 items). While brief tools are practical, their scientific limitations are significant. A complex concept like lifestyle habits cannot be fully captured with a few questions. However, as a first-step initial screening in a very busy setting, short tools can stimulate a conversation between the clinician and patient/family and create broad awareness of a health issue.

## 4. Conclusions

The timely identification of childhood health risk factors, extending beyond body mass index alone, necessitates early screening for physical inactivity, poor nutrition, inadequate sleep, and chronic stress. The OWL screening tool addresses this need through a holistic approach that integrates psychological evaluation factors, evidence-based recommendations from major health societies, and the validated strengths of existing instruments in pediatric research. However, the tool’s ability to accurately identify and stratify lifestyle risk factors has not yet been proven. Therefore, clinical application of the tool is premature. The next step should be empirical validation in future studies focusing on assessing the reliability and validity of the tool in larger, more diverse cohorts to confirm its psychometric strength.

## Figures and Tables

**Figure 1 healthcare-13-02731-f001:**
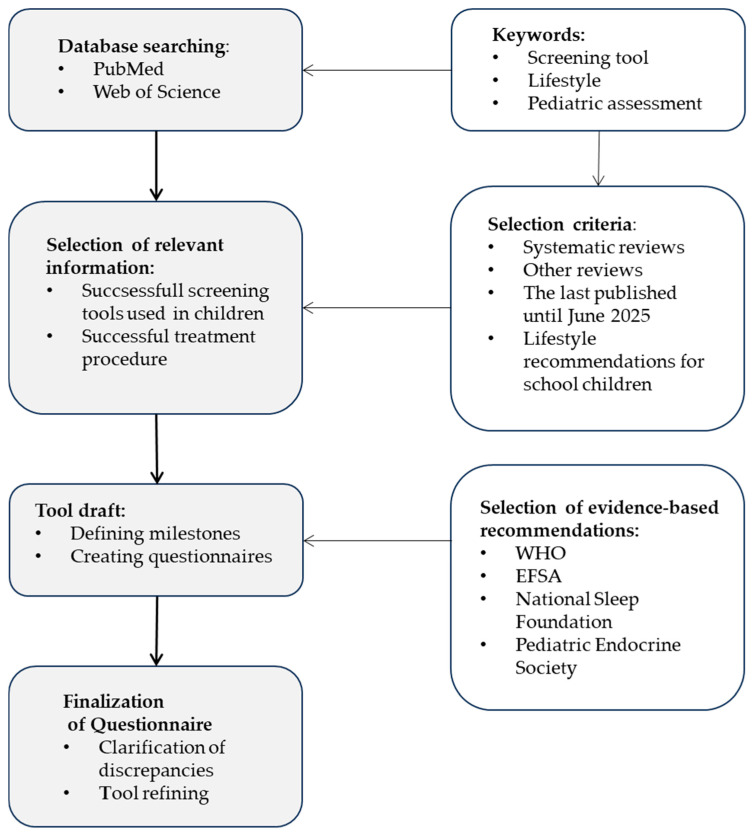
Overview of the development process of OWL Screening Tool (Optimize Wisely your Lifestyle).

**Table 1 healthcare-13-02731-t001:** Selected studies in children investigating lifestyle factors.

Name of Study or Tool/Country	Participants, Age	Design	Success of the Tool
KIDMED/Spain ^1^	*n* = 2889 6–24 years	16-item Mediterranean Diet Quality index	Link between high adherence to the Mediterranean diet and anti-inflammatory potential of the diet
ISCOLE/12 countries (Australia, Brazil, Canada, China, Colombia, Finland, India, Kenya, Portugal, South Africa, United Kingdom, United States) ^2^	n = 7372 9–11 years	Several questionnaires included questions related to food consumption, health, and well-being, and physical activity and sleep via accelerometer	Transparent model of relationships between lifestyle behaviours, environment, and obesity across countries
SACLA/France ^3^	*n* = 216 8–10 years	20-item child-reported questionnaire covering diet, activity, sleep, screen time	High reliability (78% test–retest agreement)
LE8/China ^4^	*n* = 57,210 7–17 years	Holistic approach: Combines 4 health metrics (diet, activity, nicotine exposure, BMI), healthy lifestyle education for children and parents	School-based intervention and family-based activities improved cardiovascular health by 0.89 points
COCOS/six countries (The Netherlands, Scotland, Canada, South Africa, England, and the United States) ^5^	*n* = 159 8–13 years	Superhero exercise and superpower outcome—Children’s opinions on lifestyle	Child-friendly methods including children’s voices
CLASS/Canada ^6^	*n* = 4253 10–11 years	Harvard Food Frequency Questionnaire for Youth/ Adolescents (147-item validated questionnaire); Physical Activity Questionnaire for children (10-item self-recall); questions for screen time and sleep	A holistic approach leads to greater intervention success than a single-aspect approach
NutricheQ/Ireland ^7^	*n* = 371 12–36 months	18-item questionnaire for parents, nutritional risk screening tool	Quickly assessing dietary quality (3–5 min); reliability of the test 85%
CogniDROP/Germany ^8^	*n* = 250 10–12 years	PA measurement via GT3X ActiGraph, rating of urine colour exactly water intake measurement, questions about the sleep	Higher PA improved cognition; U-Shape relation between water intake and RT of cognitive performance
Fly-Kids/Netherlands ^9^	*n* = 201 1–3 years	10-item parent-administered lifestyle screening (nutrition, PA, screen time, sleep)	Holistic approach, parental, and healthcare professionals’ satisfaction

^1^ KIDMED (Mediterranean Diet Quality Index for children and adolescents) [18]; ^2^ ISCOLE (International Study of Childhood Obesity, Lifestyle & Environment) [10]; ^3^ SACLA (Self-Administered Children’s Lifestyle Assessment) [22]; ^4^ LE8 (Life’s Essential 8, National Health Lifestyles Intervention) [1]; ^5^ COCOS (The Child Opinions on a Core Outcome Set) [2]; ^6^ CLASS Study (Children’s Lifestyle and School performance Study) [23]; ^7^ NutricheQ (Nutritional questionnaire) [24]; ^8^ CogniDROP (Cognition, Drinking Observation and Physical Activity) [20]; ^9^ Fly-Kids (Features of Lifestyle in Young Kids) [21]; *n* - number of participants.

**Table 2 healthcare-13-02731-t002:** Characteristics of the OWL Screening.

Characteristics	
Purpose	Lifestyle screening for recording health risks Guide for developing healthy lifestyle habits
Holistic approach	Looking at nutrition, physical activity, sleep, and stress management
Evaluation	Self-assessment by child and parents or assessed by interviews
Contents	Based on scientific data and guidelines from professional societies
Design	20-item questionnaire, 5 questions per lifestyle indicator
Question forms	Closed-ended questions which are answered with “yes” or “no”
Motivation and goal theory	Using approach-oriented goals and positive language
Additional features	Family structure survey questions
Target audience	School children
Scoring methods	Calculation of the score from the answers (maximum 20 points possible)
Strategies used to prevent biased	No questions regarding restrictions No multiple-choice answer No complex and long questionnaire

**Table 3 healthcare-13-02731-t003:** The template provides an example for screening schoolchildren aged 6 to 12 years—four dimensions of lifestyle factors (physical activity, nutrition, sleep, and stress management) within the context of pediatrics.

**Question**	YES	NO
**A. Nutrition**		
Do you drink about 1.2 litres of water (approx. 40 mL/kg body weight) every day?		
Do you eat whole grains every day (oatmeal, wholemeal bread and rice, quinoa, barley, wholemeal pasta, etc.)?		
Do you eat at least 2 portions of fruit and 2 to 3 portions of fresh or cooked vegetables per day? *		
Do you eat 3 meals a day with a portion of protein-rich foods (dairy products, eggs, fish, meat, legumes, tofu, nuts and seeds)? *		
Do you consume nuts regularly (at least 2–3 times per week)?		
**B. Physical activity**		
Do you reach 10,000 steps every day? (approx. 1 min—100 steps, 100 min—10,000 steps)		
Do you spend an average of 60 min per day on moderate to intensive physical activity (running, swimming, cycling, school sport)? **		
Do you train muscle strength at least three times a week (workout in school, sprint, weight training)?		
Do you play games such as football, basketball, coordination sports, dance, martial arts or yoga, etc. at least once a week?		
Do you do physical activity every day outdoors (school sport, cycling, swimming, walking, plaing active games)?		
**C. Sleep**		
Do you sleep 9–11 h a every night?		
Do you sleep mostly well at night?		
Do you go to bed at least between 9 and 10 p.m.?		
Do you have enough time to have breakfast after you wake up, before school?		
Do you practise evening rituals (walk, breathing, praying, gratitude, silence, etc.) at least an hour before going to bed?		
**D. Stress management**		
Do you enjoy a hobby every week (sport, art, gardening, cooking, sewing, singing, reading etc.)?		
Are you meeting up with friends to have fun and enjoy a good time every week?		
Do you practise relaxation techniques every day (breathing, yoga, meditation, etc)?		
Does the family regularly engage in leisure activities together (at least on weekends and public holidays)?		
Does your family have set rules for screen time?		
**Criteria for 1 point for each positive ‘yes’ answer to the question**		

Examples: recommended values of daily water intake for children aged 10–12 (44 mL/kg for boys and 39 mL/kg for girls) [31]; The WHO recommendations on physical activity cover the age group of children from 5 to 17 years for girls and boys equally [11]; Recommended sleep duration for children aged 6–12 are 9–11 h per night without sex differences [36]; * One portion corresponds to a handful; ** A single activity or several shorter activities throughout the day.

## Data Availability

The original contributions presented in this study are included in the article. Further inquiries can be directed to the corresponding author.

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
