# Peer review of "The OWL Screening Tool—A Protocol for Holistic Pediatric Lifestyle Assessment"

_healthcare, 2025, doi:10.3390/healthcare13212731_

Round 1

Reviewer 1 Report

Comments and Suggestions for Authors

Review Report

Review

The OWL Screening Tool – A Novel Instrument for Holistic Pediatrics Lifestyle Assessment

This study is compelling in its attempt to develop the OWL instrument by integrating evidence-based recommendations from major health organizations, incorporating psychological principles, and adapting validated components from existing pediatric screening tools. The design prioritizes flexibility, making it potentially suitable across different age groups and settings. However, several important points require clarification or strengthening.

-------------------------------------------------------------------

  1. Clarity of Expression

The sentence:

“The identification of childhood health risk factors must extend 9 beyond body mass index to include modifiable lifestyle behaviors”

is confusing. It is suggested to rephrase as:

“The identification of health risk factors in children should go beyond relying solely on body mass index and also include lifestyle behaviors that can be modified.”

This revision improves clarity and grammatical flow.

-------------------------------------------------------------------

  1. Figure 1 Attribution

It should be clarified whether Figure 1 is original (created by the authors) or adapted from an existing source.

If adapted, the source reference must be explicitly provided to maintain academic transparency and avoid issues of plagiarism.

-------------------------------------------------------------------

  1. Validity and Reliability

The authors mention that the instrument was tested for validity and reliability, but do not suffciently detail the methods used or present clear outcomes.

Key methodological questions include:

  • Did they employ Exploratory Factor Analysis (EFA), Confirmatory Factor Analysis (CFA), Multitrait–Multimethod (MTMM) matrix, or another approach?

It is recommended that the authors provide:

  1. A description of the validation design, including aspects such as:
  • Construct validity
  • Convergent and discriminant validity
  • Content and response process validity
  1. Statistical evidence, such as:
  • Factor loadings and model fit indices from SEM
  • Inter-construct correlations
  • Average Variance Extracted (AVE)
  • Internal consistency reliability
  • Test–retest reliability
  1. Variance decomposition (if MTMM or SEM with method factors is used), distinguishing between variance due to trait and method.

Without such evidence, the interpretability and trustworthiness of the instrument remain questionable.

-------------------------------------------------------------------

  1. Threshold or Cut-Off Definition

The manuscript lacks discussion of an optimal threshold or cut-off point.

What minimum score indicates a potential health risk in children as assessed by OWL? The authors should explain how cut-off points were determined, possibly through: - ROC curve analysis,

- Sensitivity–specificity trade-offs, or

- Empirical/clinical criteria.

Author Response

Response to Reviewer 1 Comments

This study is compelling in its attempt to develop the OWL instrument by integrating evidence-based recommendations from major health organizations, incorporating psychological principles, and adapting validated components from existing pediatric screening tools. The design prioritizes flexibility, making it potentially suitable across different age groups and settings. However, several important points require clarification or strengthening.

We thank the reviewer for the evaluation of our paper and for the constructive critique, comments, and suggestions. We greatly appreciate the time the reviewer has spent optimizing our manuscript. We considered all issues addressed and revised our paper accordingly.

  1. Clarity of Expression

The sentence: “The identification of childhood health risk factors must extend 9 beyond body mass index to include modifiable lifestyle behaviors” is confusing.

It is suggested to rephrase as:

“The identification of health risk factors in children should go beyond relying solely on body mass index and also include lifestyle behaviors that can be modified.” This revision improves clarity and grammatical flow.

Thank you for this comment and suggestion for improvement. We have adjusted the sentence accordingly and formulated it more succinctly.

“The identification of health risk factors in children should rely not only on body mass index but also on modifiable lifestyle behaviors.”

-------------------------------------------------------------------

  1. Figure 1 Attribution

It should be clarified whether Figure 1 is original (created by the authors) or adapted from an existing source.

If adapted, the source reference must be explicitly provided to maintain academic transparency and avoid issues of plagiarism.

We are very grateful to the reviewer for this valuable information. Figure 1 was designed by the authors and created with the help of external graphic designers. We refer to this in the acknowledgements. As another reviewer suggested removing the figure from the manuscript and using it as a graphical abstract instead, we will use Figure 1 for this purpose after consulting with the editorial team and will transparently cite it as our own representation.

-------------------------------------------------------------------

  1. Validity and Reliability

The authors mention that the instrument was tested for validity and reliability, but do not suffciently detail the methods used or present clear outcomes.

We are grateful for this comment. It shows us that the transparency of the goal was not clearly presented. Line 91: This paper describes the development of this tool. No validation was carried out, only a concept description. We also point this out in the ‘Methods’ section, discuss it, and explain the lack of validation in the ‘Limitations’ section.

Since other authors also suggested more references to the concept of the article in the abstract and title, we have made corresponding changes to the title and abstract. We have also placed greater emphasis on the section Methods for conceptualization to increase the transparency of the objective. 

Key methodological questions include:

  • Did they employ Exploratory Factor Analysis (EFA), Confirmatory Factor Analysis (CFA), Multitrait–Multimethod (MTMM) matrix, or another approach?

As mentioned above, we have not conducted any empirical investigations, but will take the proposed tests and calculations into account in the planned study. To provide greater transparency regarding the methodological approach, we have referred more strongly to the ‘proof of concept’ throughout the manuscript.

It is recommended that the authors provide:

  1. A description of the validation design, including aspects such as:
  • Construct validity
  • Convergent and discriminant validity
  • Content and response process validity
  1. Statistical evidence, such as:
  • Factor loadings and model fit indices from SEM
  • Inter-construct correlations
  • Average Variance Extracted (AVE)
  • Internal consistency reliability
  • Test–retest reliability
  1. Variance decomposition (if MTMM or SEM with method factors is used), distinguishing between variance due to trait and method.

Without such evidence, the interpretability and trustworthiness of the instrument remain questionable.

We would also like to thank the reviewer for all methodological and statistical comments on validation, which will be considered in the planned study in the future (Line 94-98). As other reviewers had suggested presenting the article as a developmental framework, we added the section ‘Results’ to the section ‘Methods, in line with a proposed concept for the planned validation of the OWL instrument.                                  

-------------------------------------------------------------------

  1. Threshold or Cut-Off Definition

The manuscript lacks discussion of an optimal threshold or cut-off point.

We agree with the reviewer. Unfortunately, we do not have the data to justify this. We are also aware and regret that we cannot meet the reviewer's expectations.

What minimum score indicates a potential health risk in children as assessed by OWL? The authors should explain how cut-off points were determined, possibly through: - ROC curve analysis,

- Sensitivity–specificity trade-offs, or

- Empirical/clinical criteria.

We also agree with the reviewer that the lack of validations and statistical tests does not provide evidence for the proposed instrument. Therefore, we cannot discuss missing measurements. Following the discussion, we have emphasized the section on ‘Limitations’ more strongly. We have also highlighted corresponding additions regarding the lack of empirical tests in the ‘Methods’ section, as recommended by other reviewers.     

Reviewer 2 Report

Comments and Suggestions for Authors

This is an interesting and timely paper. The topic is relevant and well aligned with current health-promotion priorities. The following areas require clarification and methodological strengthening before publication:

1)The manuscript presents OWL as a screening instrument but does not describe any formal validation process (expert panel, pilot testing, reliability, or construct validity). Please reframe this as a development and validation protocol and outline the next steps (content validity, cognitive interviews, field testing, and psychometric analysis). The title of this work should reflect this change in perspective.

2)The exclusive use of yes/no items may limit variability and sensitivity. Please justify this choice and indicate whether graded response options or weighted items were considered.

3)Cut-points for several items (e.g., steps/day, sleep duration, hydration) vary by age and sex but are presented generically. It would strengthen the paper to specify the exact normative thresholds and their sources.

4)To further strengthen the rationale, you could frame OWL within the broader trend of increasing medical and nursing complexity among pediatric patients, which underscores the need for early, multidimensional screening tools capable of identifying risk trajectories before they translate into higher care complexity. This perspective is well reflected in recent studies (e.g., PMID: 39857934), showing that greater medical complexity is closely linked to higher nursing care complexity in hospitalized children. Positioning OWL as an upstream preventive instrument within this evolving scenario would substantially enhance its relevance and long-term impact.

5) Please, improve the abstract and particularly create a strong background.

6) Clarify how the tool will be linguistically and culturally adapted for use across different contexts and socioeconomic groups within the UAE.

Author Response

Response to Reviewer 2 Comments

This is an interesting and timely paper. The topic is relevant and well aligned with current health-promotion priorities. The following areas require clarification and methodological strengthening before publication:

We thank the reviewer for the thorough evaluation of our paper and for the constructive critique and suggestions. We greatly appreciate the time the reviewer has spent optimizing our manuscript. We considered all issues addressed and revised our paper accordingly.

1)The manuscript presents OWL as a screening instrument but does not describe any formal validation process (expert panel, pilot testing, reliability, or construct validity). Please reframe this as a development and validation protocol and outline the next steps (content validity, cognitive interviews, field testing, and psychometric analysis). The title of this work should reflect this change in perspective.

We thank the reviewer for this important point. We also agree that the lack of validation of the tool leads to insufficient information. We have therefore reworded our work as a development protocol and outlined the next steps in the ‘method' section. The title and abstract have also been adjusted to reflect this change. The conclusions also include the changes.                          

2)The exclusive use of yes/no items may limit variability and sensitivity. Please justify this choice and indicate whether graded response options or weighted items were considered.

We agree with the reviewer that the justification for the choice of question format requires further explanation. We therefore explained our choice of question format based on the PREDIMED study (in the Method section 2.1. OWL – Screening). Compared to the FLY Kids questionnaire with graded response options, OWL Screening was developed based on the PREDIMED study, which did not include graded responses.

3)Cut-points for several items (e.g., steps/day, sleep duration, hydration) vary by age and sex but are presented generically. It would strengthen the paper to specify the exact normative thresholds and their sources.

Thank you for your interesting comment. Following the reviewer's recommendations, we have included the relevant recommendations on sex and age in the legend below the screening template. As another reviewer recommended presenting the figure 2 with the OWL screening questionnaire as a table and removing the icons, the entire template is now displayed as a table.

4)To further strengthen the rationale, you could frame OWL within the broader trend of increasing medical and nursing complexity among pediatric patients, which underscores the need for early, multidimensional screening tools capable of identifying risk trajectories before they translate into higher care complexity. This perspective is well reflected in recent studies (e.g., PMID: 39857934), showing that greater medical complexity is closely linked to higher nursing care complexity in hospitalized children. Positioning OWL as an upstream preventive instrument within this evolving scenario would substantially enhance its relevance and long-term impact.

We would like to thank you for the constructive and helpful comments. We also believe that publication mentioned is highly suitable for emphasizing the importance of OWL screening. We have therefore strengthened the argumentation for the OWL screening instrument on the basis of the literature cited at the end of the discussion.  It states: The trend of increasing pediatric care complexity underscores the need for early, multidimensional tools like the OWL screener to identify risk trajectories before they lead to higher care dependency. Recent research confirms this, showing that medical complexity (measured by DRG weight) significantly correlates with nursing care complexity in hospitalized children [51] By identifying modifiable lifestyle risks upstream, the OWL tool acts as a preventive instrument to mitigate this trajectory, enhancing its long-term impact.

5) Please, improve the abstract and particularly create a strong background.

Regarding the reviewer's recommendations, we have corrected the abstract and supporting information. All changes are highlighted in yellow in the manuscript.

6) Clarify how the tool will be linguistically and culturally adapted for use across different contexts and socioeconomic groups within the UAE.

We are very grateful for this important and interesting comment. There are indeed plans to produce the questionnaire in the two official languages of the UAE, namely English and Arabic. This is to ensure that it is linguistically adapted for use in the UAE. Cultural differences will be considered in the next step, namely the validation of the screening instrument.

We have supplemented and highlighted this interesting aspect in the method section.

Reviewer 3 Report

Comments and Suggestions for Authors

This manuscript presents the OWL tool designed to assess lifestyle risk factors in children aged 6–12. The authors integrate existing validated questionnaires and professional health recommendations into a dichotomous format. However, the paper presents only conceptual development, without empirical testing.

The main research question is clearly stated and relevant

Holistic integration of four domains is commendable. Positive goal-framing is a fresh concept.

The claim of novelty is overstated; similar multidomain tools exist, and the integration here is largely conceptual. The scientific gap (lack of holistic pediatric tools) is asserted but not supported with a formal gap analysis or systematic comparison.

The concept is moderately original but methodologically premature for publication in a journal emphasizing empirical health research.

Line 14. “integrating evidence-based recommendations…” — specify which societies (WHO….).

Systematic literature basis and comparison with 9 existing tools. Clear alignment with WHO and other recommendations.

No validation, no pilot testing, or psychometric analysis, making it descriptive, not analytical. The study should include psychometric testing (pilot study with n > 100 children), provide reliability and factor analysis…

Methodological transparency is partial: selection criteria for questions are described narratively but not systematically, no inclusion/exclusion rules… Lacks a flow diagram or process illustrating item generation and selection. Clarify literature search method (databases, keywords).

Table 1. Line 111 lacks consistent citation formatting.

Results section only restates design features; no data presented.

Discussion.  statements like “potential to improve long-term outcomes” are speculative. Repetition of introduction content (especially regarding Mediterranean diet and family influence).

Limitations section acknowledges lack of validation but minimizes its significance.

Language is generally fluent and professional. Logical flow between sections.

L310 “compered” → “compared”

Figure 1 is visually beautiful but does not contain details to explain the four domains. Consider submitting it separately as Graphical Abstract.

Figure 2. Consider naming it as a Table. The icons do not contain any scientific information and could be removed.

It has potential utility and implications, however, until validated, it remains a conceptual proposal, not a clinically applicable instrument. The authors should frame it as a developmental paper (e.g., “Phase 1: Conceptualization and Design”) rather than implying operational readiness.

Conclusion. should focus on limitations and next steps (validation, testing).

Author Response

Response to Reviewer 3 Comments

We thank the reviewer for the thorough evaluation of our paper and for the constructive critique and suggestions. We greatly appreciate the time the reviewer has spent optimizing our manuscript. We considered all issues addressed and revised our paper accordingly.

This manuscript presents the OWL tool designed to assess lifestyle risk factors in children aged 6–12. The authors integrate existing validated questionnaires and professional health recommendations into a dichotomous format. However, the paper presents only conceptual development, without empirical testing.

We agree with the reviewer. Based on the comments of the other reviewers, we have added additions to the manuscript to improve transparency regarding the methodological approach and to highlight the lack of empirical evidence.

The main research question is clearly stated and relevant. Thank you very much for this comment.

Holistic integration of four domains is commendable. Positive goal-framing is a fresh concept. The claim of novelty is overstated; similar multidomain tools exist, and the integration here is largely conceptual.

Thank you for pointing this out. We agree that we did not sufficiently justify the ‘novelty’ of the OWL screening. The novelty of the OWL screening was originally intended to be justified not only by its holistic approach, but also by its design, user-friendliness and the proven absence of such an instrument in studies involving children, as summarized in the systematic review by other authors (number 5 in the manuscript).

Since we did not carry out the systematic review ourselves and were unable to sufficiently confirm the user-friendliness through empirical testing, we removed the word ‘novelty’ from all instances in the manuscript and toned down the wording accordingly.

The scientific gap (lack of holistic pediatric tools) is asserted but not supported with a formal gap analysis or systematic comparison. We greatly appreciate your feedback. We base our assumption on two reviews from 2022 and 2023, which we cite [5,12] (line 61-62). However, we have not verified the status of screening instruments for schoolchildren ourselves. For this reason, we no longer used the term ‘scientific gap’ in the manuscript as justification for our target setting.

The concept is moderately original but methodologically premature for publication in a journal emphasizing empirical health research. We greatly appreciate all reviewer comments. To improve the section on methodology, we have emphasized that the work presented is a first-phase development study focusing on conceptualization and content development. All additions have been highlighted in yellow in the text.

Line 14. “integrating evidence-based recommendations…” — specify which societies (WHO….). Following the recommendation of the reviewer, we have made the appropriate additions. We have added a new figure showing the development process.

Systematic literature basis and comparison with 9 existing tools. Clear alignment with WHO and other recommendations. To increase transparency, we have included references to the ‘evidence-based recommendations of the societies’ in the ‘Methods’ section and in Table 3 (previously Figure 2, which was converted to a table format on the recommendation of the reviewer).

No validation, no pilot testing, or psychometric analysis, making it descriptive, not analytical. The study should include psychometric testing (pilot study with n > 100 children), provide reliability and factor analysis… We fully agree with the reviewer. The lack of empirical research does not allow for analytical evidence and discussion. We are aware of this and regret that we are not yet able to provide such data. On the recommendation of other reviewers, we have converted our manuscript into a protocol and rationale. This means that the ‘Results’ section has been omitted. The content of the results has been added to the ‘Methods’ section. We have adjusted the title in line with the protocol to make the objective clearer.

Methodological transparency is partial: selection criteria for questions are described narratively but not systematically, no inclusion/exclusion rules… Lacks a flow diagram or process illustrating item generation and selection. Clarify literature search method (databases, keywords). Thank you very much for your constructive comments, which we have considered. We have now created a flow chart that illustrates our development process with the corresponding steps.

Table 1. Line 111 lacks consistent citation formatting. Formatting has been corrected.

Results section only restates design features; no data presented. As mentioned above, we have added the ‘results’ section to the methods part.

Discussion.  statements like “potential to improve long-term outcomes” are speculative. Thank you for your opinion. We admit that we have not yet tested this and have therefore toned down the wording.

Repetition of introduction content (especially regarding Mediterranean diet and family influence). We thank and appreciate of the reviewer attention. We have removed our redundancy regarding family and the Mediterranean diet from the discussion.

Limitations section acknowledges lack of validation but minimizes its significance. We agree with the reviewer. To further emphasize the significance of the limiting factors, we have added supplementary explanations to the ‘Limitations’.

Language is generally fluent and professional. Logical flow between sections. Thank you for this comment.

L310 “compered” → “compared” Has been corrected.

Figure 1 is visually beautiful but does not contain details to explain the four domains. Consider submitting it separately as Graphical Abstract. On your recommendation, we have removed Figure 1 and are preparing the graphical abstract.

Figure 2. Consider naming it as a Table. The icons do not contain any scientific information and could be removed. Following your suggestion, we have reformatted Figure 2 into a table.

It has potential utility and implications, however, until validated, it remains a conceptual proposal, not a clinically applicable instrument. The authors should frame it as a developmental paper (e.g., “Phase 1: Conceptualization and Design”) rather than implying operational readiness. We fully agree with the reviewer. It was not our intention to give this impression. We have therefore emphasized all references to the concept presentation and the rationale more strongly.

Conclusion. should focus on limitations and next steps (validation, testing). Thank you very much for this suggestion for improvement. We have made the appropriate changes.

Round 2

Reviewer 1 Report

Comments and Suggestions for Authors

The authors have revised the manuscript in accordance with the reviewers’ suggestions. I conclude that the manuscript can be accepted in its current form.

Reviewer 2 Report

Comments and Suggestions for Authors

I have carefully reviewed the revised version of the manuscript and the point-by-point responses. The revision successfully addresses all comments raised by the reviewer, resulting in a clearer, more robust, and well-structured manuscript. Well done, and congratulations on the quality of the revision.

Reviewer 3 Report

Comments and Suggestions for Authors

The authors responded adequately to my comments. I have no further comments.